# Interface Structure, Dielectric Behavior and Temperature Stability of Ba(Mg_1/3_Ta_2/3_)O_3_/PbZr_0.52_Ti_0.48_O_3_ Thin Films

**DOI:** 10.3390/ma16196358

**Published:** 2023-09-22

**Authors:** Zhi Wu, Yifei Liu, Jing Zhou, Hong Zhao, Zhihui Qin

**Affiliations:** 1School of Materials and Engineering, Hunan Institute of Technology, Hengyang 421002, China; wuzhi0549@163.com (Z.W.); liuyifei0213@whut.edu.cn (Y.L.); zhaohong165@163.com (H.Z.); zhu0764zheng@163.com (Z.Q.); 2State Key Laboratory of Advanced Technology for Materials Synthesis and Processing, School of Materials Science and Engineering, Wuhan University of Technology, Wuhan 430070, China

**Keywords:** BMT/PZT thin films, interface number, dielectric behavior, temperature stability

## Abstract

Multilayer films can achieve advanced properties and a wide range of applications. The heterogeneous interface plays an important role in the performances of multilayer films. In this paper, the effects of the interface of Ba(Mg_1/3_Ta_2/3_)O_3_/PbZr_0.52_Ti_0.48_O_3_ (BMT/PZT) thin films on dielectric behavior and temperature stability are investigated. The heterogeneous interface structures are characterized by Auger electron spectroscopy (AES). The PZT-BMT interface is different from the BMT-PZT interface in thickness. For the PZT-BMT interface, the PZT thin films are diffused to the whole BMT layers, and the interface thickness is about 90 nm, while the BMT-PZT interface’s thickness is only about 8.6 nm. The presence of heterogeneous interfaces can improve the performances of BMT/PZT thin films and expand their applications. The dielectric constant of BBPP thin films is significantly lower than BPBP thin films, while the dielectric loss is exactly the opposite. The more interfaces there are, the greater the dielectric constant. The relationship between the electric-field-dependent dielectric constant curve and the P-E curve is established. The equivalent interface barrier of the diode is used to show that the dielectric peaks under the positive and negative voltage are different. Similarly, heterogeneous interfaces show a certain improvement in dielectric tunability and temperature stability.

## 1. Introduction

PbZr_x_Ti_1−x_O_3_ (PZT) thin films can be used extensively in capacitors, filters, energy harvests, dynamic random access memories (DRAMs) and acoustic surface wave devices due to their good electrical properties [1,2,3,4,5,6]. However, their reliability and temperature sensitivity limit the applications of PZT thin films. Ba(Mg_1/3_Ta_2/3_)O_3_ (BMT) thin films with high quality factor and good temperature stability have received intensive attention [7,8]. The advanced properties may be realized by heterogeneous lamination, which is fabricated by combining BMT thin films and PZT thin films.

Compared with homogeneous thin films, multilayer thin films have higher quality factor. The structure of multilayer thin films plays a key role in their dielectric loss. Lv et al. [9] asserted that the dielectric constant of (Na_0.8_K_0.2_)_0.5_Bi_0.5_TiO_3_/Ba_0.5_Sr_0.5_(Ti_0.97_Mn_0.03_)O_3_ thin films showed a growth trend with the increase in periodic number. The dielectric loss of BiFeO_3_/BaTiO_3_ thin films was gradually reduced with the increase in interface number increase due to the stress at the interface region [10]. Liu et al. [11] asserted that the variable Bi_0.96_Sr_0.04_Fe_0.98_Co_0.02_O_3_ (BSFCO) thickness had an impact on the ferroelectric and dielectric behavior of BSFCO/CoFe_2_O_4_ thin films and the dielectric loss varied with the BSFCO thickness. Researchers used the Maxwell–Wagner effect to explain that the dielectric constant increased with the interface number [12,13]. The conductivity and dielectric constants of different dielectric materials and interfaces showed variations. The charge accumulated at the interface of the dielectric thin film under the electric field, increasing the conductivity and polarization of the thin films.

The heterogeneous lamination of multiple films can also improve the temperature coefficient. The (Pb_0.92_La_0.08_)(Zr_0.65_Ti_0.35_)O_3_/PbZrO_3_ thin films with appropriate film structure showed excellent temperature stability and fatigue endurance [14]. Lee et al. [15] asserted that the stacking sequence affected the dielectric behavior of MgTiO_3_/CaTiO_3_ (MT/CT) thin films and the temperature stability was controlled by adjusting the thickness of the CT thin film. Song et al. [16] studied Bi_0.2_Sr_0.7_TiO_3_/BiFeO_3_ (BST/BFO) thin films and found that the interfacial effect and multiple structures could enhance the polarization and breakdown strength of BST/BFO thin films that also had stable energy storage performance at 35–140 °C. Zhou et al. [17] found that the introduction of the Ti layer could improve the performance of the Sb_70_Se_30_/Ti thin films via a smoother surface and smaller volume changes, and the Sb_70_Se_30_/Ti thin films had better interface performance and stability.

The heterogeneous lamination of multiple films may introduce new properties. Nan et al. [18,19] combined ferroelectric thin films with magnetic thin films, and the heterogeneous lamination of multiple films produced a magnetoelectric effect that neither thin film possessed alone. The Raman scattering was used to observe the magnetoelectric coupling effect of the multiple films, and the strain generated by the CoFe_2_O_4_ (CFO) thin films led to the shift of the Raman scattering peak of the PZT piezoelectric thin films, proving that the strain resulted in magnetoelectric coupling of thin films [20]. BiFeO_3_/BaTiO_3_ (BFO/BTO) thin films combined the advantages of BFO and BTO thin films, with a lower tunneling current and stronger magnetoelectric effect [21]. Burdin et al. [22] demonstrated theoretically and experimentally that the ferromagnetic layer had a great influence on the magnetoelectric effect of the composite thin films, and the nonlinear magnetization of the magnetoelectric layer contributed an additional 14% of the magnetoelectric effect.

However, there are few studies on dielectric/ferroelectric thin films. The above-mentioned laws of multiple films may also be adapted to dielectric/ferroelectric thin films. In order to expand the applications of ferroelectric thin films, the relationship between the structures and properties of dielectric/ferroelectric thin films should be studied. In our previous work [23], we have focused on the PbZr_0.52_Ti_0.48_O_3_/Ba(Mg_1/3_Ta_2/3_)O_3_ (PZT/BMT) thin films and the interface polarization model has been set up to explain the relationship between the total dielectric constant of PZT/BMT thin films, interface number and dielectric constant of interface. In fact, the stacking sequence has an effect on the electrical properties. Zou et al. [24] focused on the heterogeneous ferroelectric thin films with different components of Pb_1.1_Zr_0.52_Ti_0.48_O_3_, Pb1.1Zr_0.2_Ti_0.8_O_3_ and Pb_1.1_Zr_0.8_Ti_0.2_O_3_. When the stacking sequence was different, the strain was different, resulting in a different hysteresis loop, a built-in electric field and the motion of the domain wall. Zhou et al. [25] showed that there was an obvious transition layer at the interface of Ca(Mg_1/3_Ta_2/3_)O_3_/CaTiO_3_ and the stacking sequence had a significant effect on the dielectric properties.

In this work, we focus on the BMT/PZT thin films and explore the influence of the interface on the dielectric behavior, dielectric tunability and temperature stability of thin films. The interface between the BMT layer and PZT layer is detected by Auger electron spectroscopy (AES) to explain the changes in the performance of heterogeneous thin films. The relationship between the electric-field-dependent dielectric constant curve and the P-E curve, and the asymmetry of the electric-field-dependent dielectric constant curve are also discussed.

## 2. Materials and Methods

### 2.1. Raw Materials

Lead acetate trihydrate (≥99.5%, China National Pharmaceutical Group Chemical Reagent Co., Ltd., Shanghai, China), zirconium n-propanol (70%, Shanghai Jingchun Biochemical Technology Co., Ltd., Shanghai, China), titanium isopropanol (≥98%, Acros Corporation, Livonia, MI, USA), ethylene glycol methyl ether (≥99.5%, China National Pharmaceutical Group Chemical Reagent Co., Ltd.) and acetylacetone (≥99.5%, China National Pharmaceutical Group Chemical Reagent Co., Ltd.) were used to prepare PZT thin films.

Tantalum pentoxide (≥99.0%, China National Pharmaceutical Group Chemical Reagent Co., Ltd.), barium carbonate (≥99.0%, China National Pharmaceutical Group Chemical Reagent Co., Ltd.), basic magnesium carbonate (≥99.0%, Silian Chemical Plant Co., Ltd., Shanghai, China), citric acid (≥99.5%, China National Pharmaceutical Group Chemical Reagent Co., Ltd.), potassium hydroxide (≥85.0%, Silian Chemical Plant Co., Ltd.), hydrochloric acid (36–38%, Kaifeng Dongda Chemical Co., Ltd., Kai Feng, China, reagent factory) and ammonia liquor (≥30%, China National Pharmaceutical Group Chemical Reagent Co., Ltd.) were used to prepare BMT thin films.

### 2.2. Preparation of Precursor Solution

#### 2.2.1. Preparation of PZT Precursor Solution

The precursor solution of PZT was prepared with the sol-gel method [26]. Since zirconium n-propanol and titanium isopropanol easily reacted with water, the crystal water in the lead acetate trihydrate should be removed before the reaction. On the other hand, high-temperature annealing can cause lead to evaporate and form lead vacancies, which can easily form a pyrochlore phase and reduce the performance of PZT thin films. Therefore, an additional 15% mole of lead was added during the preparation processes. The preparation processes of the PZT precursor solution are shown in Figure 1.

#### 2.2.2. Preparation of BMT Precursor Solution

The precursor solution of BMT was obtained using the aqueous solution gel method. Tantalum pentoxide is difficult to dissolve in other solutions, so it is necessary to prepare a clarified and transparent tantalum (tantalum peroxide citrate, P-Ta-CA) solution first. The preparation processes of the P-Ta-CA solution are shown in Figure 2.

According to the stoichiometric ratio, barium carbonate and basic magnesium carbonate were added to the P-Ta-CA solution, and then heated and stirred at 40 °C until the solution became clear. Then, the pH value of the solution was adjusted to 7–8 with NH_3_·H_2_O, and the solution was heated and stirred at 40 °C for 2 h. Finally, the clarified and transparent BMT precursor solution was obtained.

### 2.3. Preparation of BMT/PZT Thin Films

The BMT/PZT thin films, which were named PPBB, BBPP and BPBP, were obtained by the spin coating method according to Figure 3. The precursor solution was spun onto the substrate in a stacking sequence. The thin films were heated via the layer-by-layer thermal annealing method. Each PZT layer was treated at 150 °C and 350 °C for 5 min to remove solvents and organic matter. Then, the film was heated at 600 °C for 5 min to pre-crystallize. Each BMT layer was treated at 180 °C and 380 °C for 2 min to remove water and citric acid. Then, the film was heated at 600 °C for 10 min to pre-crystallize. When BMT/PZT thin films met the design requirements, the BMT/PZT thin films were annealed at 700 °C for 1 h to crystallize.

### 2.4. Characterization

The Grazing Incident XRD (PANalytical X’Pert PRO) with Cu Kα (λ = 1.5406 Å) radiation was used to detect the crystalline structures of thin films [27,28]. The FE-SEM (ULTRA Plus-43-13) was used to investigate the cross section of BMT/PZT thin films. The depth profiles of chemical composition were measured by auger electron spectroscopy (AES, model PHI 700). The Agilent HP4294A impedance analyzer was used to obtain the dielectric behavior in the frequency range from 40 Hz to 1 MHz, the variation of dielectric behavior with temperature and the electric-field-dependent dielectric constant for BMT/PZT thin films at 1 kHz. A precision workstation (Radiant Technologies Inc., Albuquerque, NM, USA) was used to measure the P-E hysteresis loop of PBPB thin films at 10 Hz.

## 3. Results and Discussion

### 3.1. Structure and Morphology Analysis

The X-ray diffraction patterns of BMT thin films, PZT thin films and BMT/PZT thin films are depicted in Figure 4. A pure-phase thin film with good crystallization was synthesized. The diffraction peaks near 2θ = 31.20°, 38.40°, 44.56°, 50.13°, 55.36°, and 64.78° were attributed to reflections with the crystallites of the BMT cubic phase [PDF files 087-1733] and PZT rhombohedral phase [PDF files 073-2022]. The diffraction peak near 2θ = 21.91° belonged to the PZT rhombohedral phase. Except for the substrate diffraction peaks, other peaks belonged to BMT or PZT thin films. The diffraction peaks in BMT/PZT thin films are hard to differentiate, except that the (100) peak is the characteristic peak of PZT thin films, since the diffraction peaks are very close to each other.

The FE-SEM observations of the cross-section of BBPP and BPBP thin films are depicted in Figure 5. In Figure 5a, it can be seen that all the thin films have good crystallization, few pores and dense growth. The BMT layer and PZT layer thicknesses are approximately 40 nm and 648 nm, respectively. According to Figure 5b, the PZT thin films have good crystallization and dense growth, with a thickness of approximately 160 nm for each PZT layer. The BMT layer could not be directly observed, since each layer is too thin to detect. However, the thickness of the BMT layer can be adjusted through the preparation process.

In order to determine the thickness of the interface, the AES depth profiles of chemical composition were measured. Figure 6 and Figure 7 show the cross-sectional AES depth profiles of PPBB and BBPP thin films, respectively. The AES depth profiles reveal the relationship between chemical composition and the thickness of interface. As shown in Figure 6, when BMT is grown on the substrate, Ba, Mg and Ta atoms diffuse into the substrate. The thickness of the interface between BMT and the substrate was about 40 nm. PZT diffused into the whole BMT layer and the thickness of the PZT-BMT interface is about 90 nm. In Figure 7a, The thickness of interface between PZT and substrate was about 110 nm, which was much thicker than the interface between BMT and the substrate. BMT thin films can modify the interface between PZT and the substrate. In order to characterize the BMT-PZT interface, the BBPP thin films were slowly swept at the first depth of 80 nm, and the results are depicted in Figure 7b. The BMT layer and PZT layer diffused each other to form an interface layer with a thickness of 8.6 nm, which is much thinner than the PZT-BMT interface. There were holes on the surface of the BMT layer [29], and the PZT precursor solution penetrated into the BMT layer to form a thick PZT-BMT interface during lamination. However, the PZT layer grew densely, and formed a BMT-PZT interface mainly in the form of diffusion.

### 3.2. Dielectric Behavior Analysis

The dielectric behaviors of BBPP and BPBP thin films are depicted in Figure 8. The dielectric constant decreased rapidly at first and then stabilized with frequencies. However, the dielectric loss first rapidly decreased and then slowly increased. The dielectric constant of BBPP thin films was lower than BPBP thin films, while the dielectric loss was exactly opposite. Without considering the interface, the dielectric constant was only related to the composition and the amount of each composition. The dielectric behaviors of BBPP and BPBP thin films should be the same. However, the dielectric behaviors of BBPP and BPBP thin films were different, indicating that the heterogeneous interface has a great impact on the dielectric behavior. The more interfaces there were, the greater the dielectric constant. The interface polarizability increased with interface number, which is beneficial to the dielectric constant [23]. According to the references [25,30], the dielectric constant decreases with decreasing interface number for the same type of microwave dielectric materials and this conclusion is also true for different types of materials.

### 3.3. Dielectric Tunability

The electric-field-dependent dielectric constant for BBPP and BPBP thin films is depicted in Figure 9. As the electric field increased, the dielectric constant first increased and then decreased. There was a dielectric peak under the appropriate electric field, and the dielectric peaks under the positive and negative voltage were different. According to the formula of dielectric tunability [31], the dielectric tunability of BBPP and BPBP thin films were 20.95% and 28.54%, respectively. Dielectric tunability can be improved by increasing the number of interfaces.

The P-E loop of BPBP thin films at 10 Hz was derived to better understand the electric field-dependent dielectric constant, and the result is shown in Figure 10. When the electric field increased to the coercive field, the change rate of polarization gradually increased. When the electric field continued to increase, the change rate of polarization gradually decreased. The relation between the dielectric constant and change rate of polarization can be expressed by Formula (1)
(1)$$\varepsilon_{0}\varepsilon =\varepsilon_{0}+\frac{\partial P}{\partial E}$$
where *ε*_0_, *ε*, *P* and *E* are the dielectric constant of vacuum and thin films, polarization and electric field, respectively. According to Formula (1), the dielectric constant is linearly related to the change rate of polarization. Therefore, there is a dielectric peak at an appropriate electric field.

The equivalent interface barrier of the diode (as shown in Figure 11) was used to demonstrate that the dielectric peak under the positive and negative voltage is different. The film–electrode interface not only showed capacitance and resistance, but also had a barrier. The junction capacitance C_J_ of the diode is composed of barrier capacitance C_B_ and diffusion capacitance C_D_. When the forward bias was applied, the diffusion capacitance was dominant. On the contrary, the barrier capacitance was dominant. Therefore, the dielectric peak value under the positive and negative voltage was different.

The temperature-dependent dielectric constant for BBPP and BPBP thin films at 1 MHz is depicted in Figure 12. The dielectric constant of BMT/PZT thin films increased gradually with the increase in test temperature. The dielectric temperature coefficients of thin films can be obtained from the literature [32], and the dielectric temperature coefficient of BBPP thin films is higher than BPBP thin films, which indicates that the increase in interface number is conducive to the improvement of the dielectric temperature stability of BMT/PZT thin films. It is mainly related to the interface relaxation activation energy, which increases with interface number, and the charge carriers on the interface are not easily excited [23]. Therefore, it exhibits good temperature stability.

## 4. Conclusions

The influence of interface on the dielectric behavior, dielectric tunability and temperature stability of BMT/PZT thin films were studied. The interface between PZT and BMT differs with varying stacking sequences. The heterogeneous interface number can improve the dielectric behaviors of BMT/PZT thin films, which also have good temperature stability and dielectric tunability. The dielectric constant decreases with decreasing interface number for the same type of microwave dielectric materials, which is also true for different types of materials. The presence of heterogeneous interfaces can improve the performance of thin films and expand the application of ferroelectric thin films.

## Figures and Tables

**Figure 1 materials-16-06358-f001:**
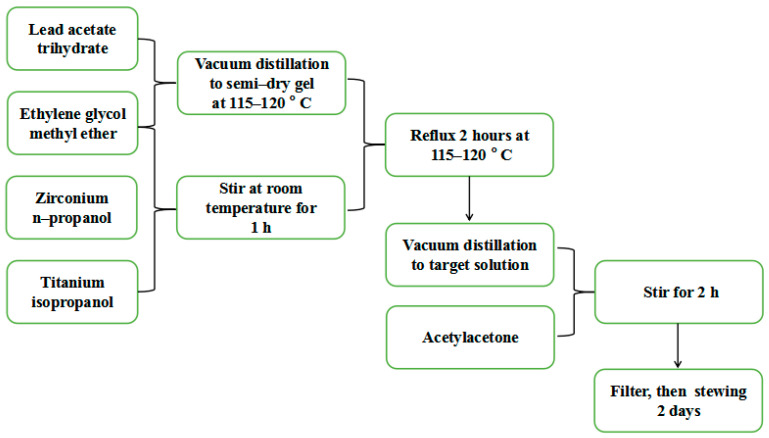
The preparation processes of PZT precursor solution.

**Figure 2 materials-16-06358-f002:**
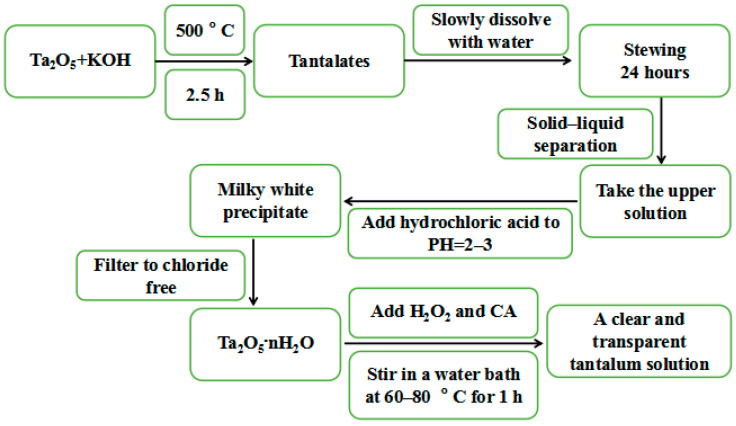
The preparation processes of P-Ta-CA solution.

**Figure 3 materials-16-06358-f003:**
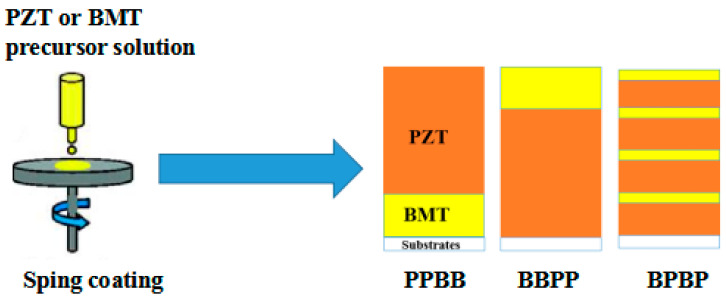
Structure design of BMT/PZT thin films.

**Figure 4 materials-16-06358-f004:**
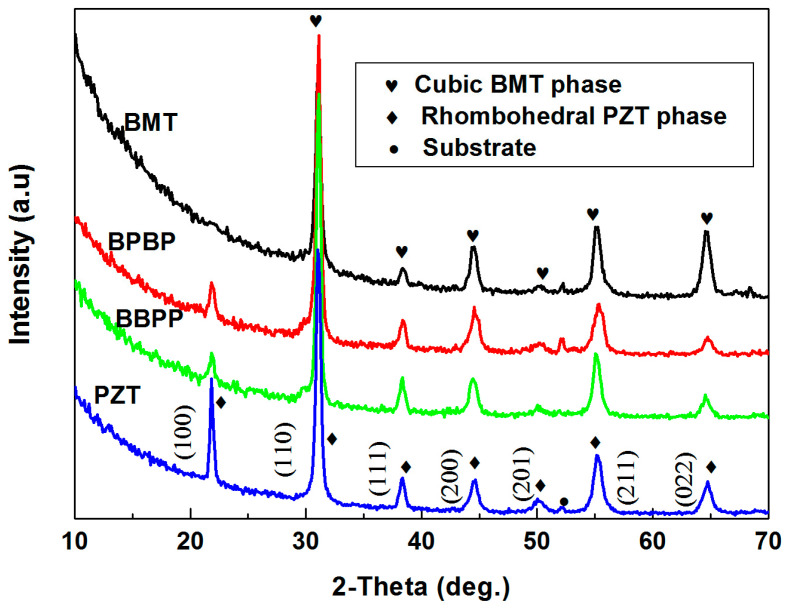
X-ray diffraction patterns of BMT thin films, PZT thin films and BMT/PZT thin films.

**Figure 5 materials-16-06358-f005:**
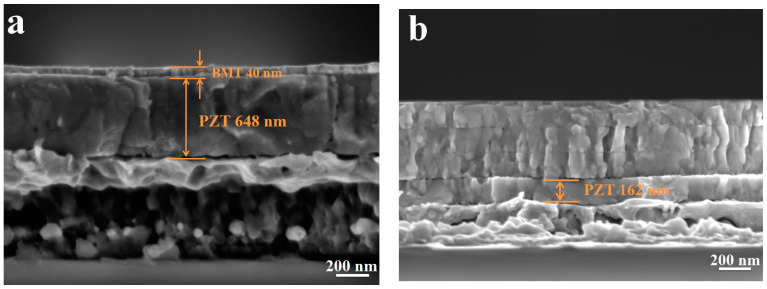
FE-SEM observations of the cross section of BBPP thin films (**a**) and BPBP thin films (**b**).

**Figure 6 materials-16-06358-f006:**
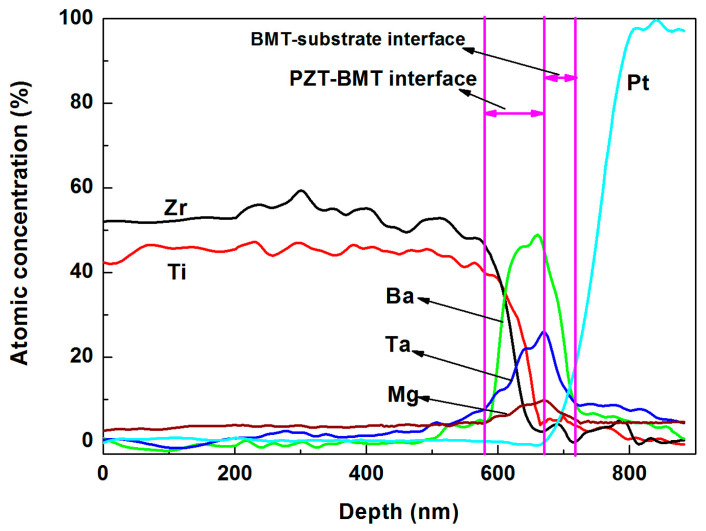
AES depth profile of PPBB thin films.

**Figure 7 materials-16-06358-f007:**
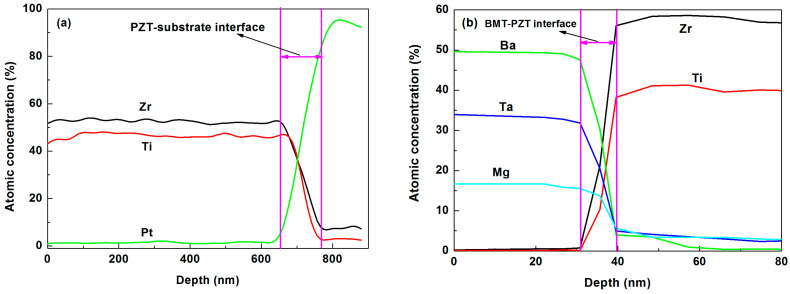
AES depth profile of BBPP thin films: full scale (**a**) and zoom scales (**b**).

**Figure 8 materials-16-06358-f008:**
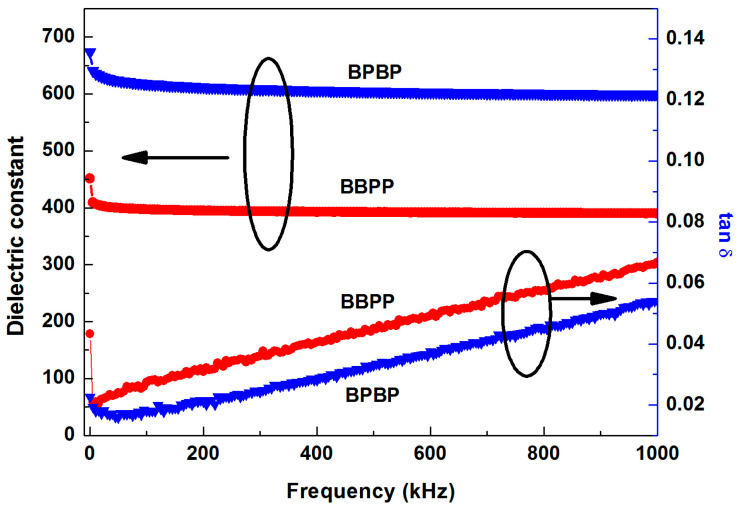
Dielectric behaviors of BBPP and BPBP thin films.

**Figure 9 materials-16-06358-f009:**
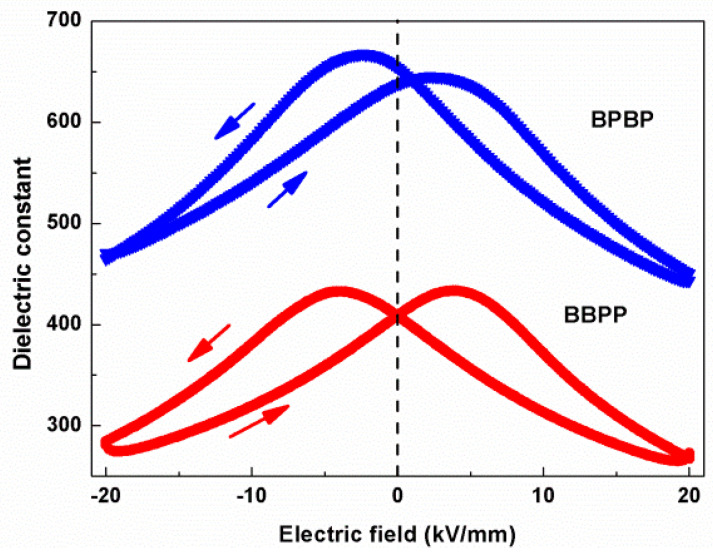
Electric-field-dependent dielectric constant for BBPP and BPBP thin films at 1 kHz.

**Figure 10 materials-16-06358-f010:**
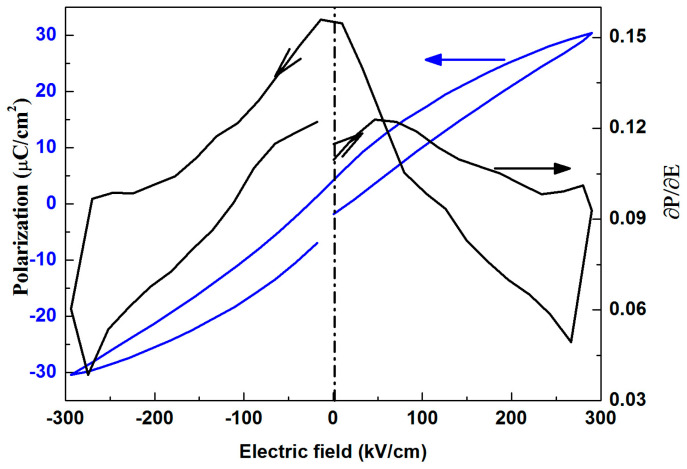
P-E loop and change rate of polarization-E loop of BPBP thin films at 10 Hz.

**Figure 11 materials-16-06358-f011:**
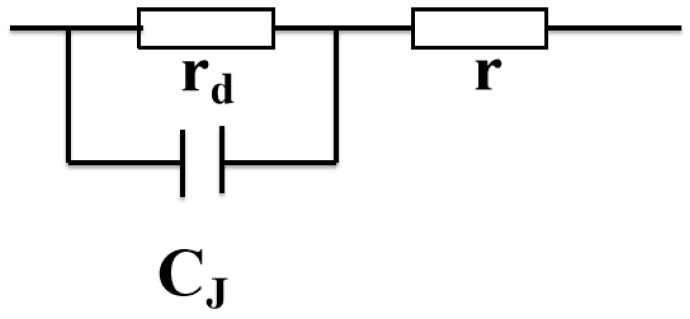
Schematic diagram of diode equivalent circuit.

**Figure 12 materials-16-06358-f012:**
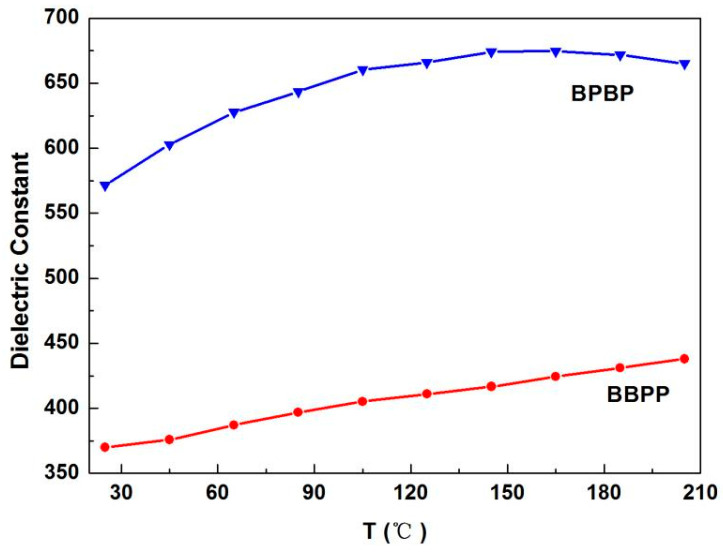
Temperature-dependent dielectric constant for BBPP and BPBP thin films at 1 MHz.

## Data Availability

Not applicable.

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
