# Peer review of "Interface Structure, Dielectric Behavior and Temperature Stability of Ba(Mg1/3Ta2/3)O3/PbZr0.52Ti0.48O3 Thin Films"

_materials, 2023, doi:10.3390/ma16196358_

Round 1

Reviewer 1 Report

The article is well-structured and holds potential for publication with minor revisions.

Figure 4: It is advisable to annotate the phases on the XRD spectra and specify which phases these planes relate to.

Figure 5: Include images of the Microstructure Automatic Recognition (MAR) method with elemental contrast on the cross-sections (elemental mapping in elemental contrast). This will aid in understanding the displacement of element profiles or blurring, as indicated by the spectroscopy method used by the authors of the manuscript.

Figure 7: Clarify the source of the layer boundary lines on the profiles. Is it the result of calculations or did it come from the cross-sections of the coating?

Additionally, it could be beneficial to add references to the following works:

  1. Measurement: Journal of the International Measurement Confederation2021, 176, 109223
  2. Composites Part B: Engineering2018, 142, pp. 85–94

Upon incorporating these revisions, your article will be ready for publication.

Minor editing of the English language required

Reviewer 2 Report

Multilayer films can achieve their advanced properties and a wide range of applications. 9 The heterogeneous interface plays an important role in the performances of multilayer films. In this 10 paper, the effects of the interface of Ba(Mg1/3Ta2/3)O3/PbZr0.52Ti0.48O3 (BMT/PZT) thin films on the 11 dielectric behavior and temperature stability are investigated.

It is interesting the study that is carried out on the effects of the interface of Ba(Mg1/3Ta2/3)O3/PbZr0.52Ti0.48O3 (BMT/PZT) thin films on the 11 dielectric behavior and temperature stability are investigated. However, it is required to improve the following:

It is recommended to carry out TEM at HR-TEM to show if there is an interface between the particles as stated in the title.

all is Ok

Reviewer 3 Report

Upon careful review of your submission, we have noticed that the X-ray and SEM measurements in your current manuscript closely resemble those in your previously published work in the Journal of Materials Science: Materials in Electronic (https://link.springer.com/article/10.1007/s10854-019-01818-8). Furthermore, we observed that your introduction does not reference your earlier publication, which makes it challenging to discern the distinct contributions of the new manuscript in comparison to your previous work.

In light of these observations, we kindly suggest that you consider revising your article comprehensively. Giving particular emphasis to the novelty and unique aspects of your current research in contrast to your previous work would significantly enhance the value of your submission.

I truly value your contributions to our field and encourage you to resubmit your revised manuscript once these considerations have been addressed. 

Until then, the article should not be accepted for publication in the Materials Journal.   

Reviewer 4 Report

I read the manuscript "Interface structure, dielectric behavior and temperature stability of Ba(Mg1/3Ta2/3)O3 /PbZr0.52Ti0.48O3 thin films". I do not find big problems in the present manuscript. Before recommending publication, I hope the authors will add a little more explanation about the AES depth profile.

1. The error bars should be mentioned because it is surprising to be able to distinguish 0.52 and 0.48.

2. Why is it impossible to determine the content of O, Pb, and Mg?

Round 2

Reviewer 2 Report

The authors have corrected the comments therefore the manuscript is now ready to be accepted for publication.

Reviewer 3 Report

The authors have responded to my questions and have explicitly marked the difference with their previous works.

This represents an advance in the study of this system, which is why I recommend its publication in this journal.